# Performance of Sweet Cherry Cultivars and Advanced Selections on Gisela 5 Rootstock in Young Orchards

**DOI:** 10.3390/plants12030614

**Published:** 2023-01-30

**Authors:** Juozas Lanauskas, Darius Kviklys, Nobertas Uselis, Vidmantas Stanys

**Affiliations:** 1Institute of Horticulture, Lithuanian Research Centre for Agriculture and Forestry, Kauno Str. 30, LT-54333 Kaunas, Lithuania; 2Department of Horticulture, Norwegian Institute of Bioeconomy Research—NIBIO Ullensvang, Ullensvangvegen 1005, NO-5781 Lofthus, Norway

**Keywords:** trunk diameter, yield, yield efficiency, fruits, soluble solids content

## Abstract

Six sweet cherry cultivars and two advanced selections of Gisela 5 rootstock were tested in 2015–2021 at the Institute of Horticulture, Lithuanian Research Centre for Agriculture and Forestry. Fruit trees were planted at distances of 4.5 × 2.5 m and trained as spindles. Orchard floor management included frequently mown grass in alleyways with herbicide strips along tree rows. Cultivars ‘Mindaugė’ and ‘Irema BS’ were the most vigorous at the end of the seventh leaf. Their trunk diameter achieved 11.6 cm. The ‘Merchant’ cultivar had the smallest trunk diameter—9.3 cm. The average yield in 2018–2021 ranged from 2.75 t/ha for ‘Vega’ to 8.73 t/ha for ‘Regina’. Cultivars ‘Regina’, ‘Sunburst’, ‘Irema BS’ and ‘Merchant’ had the highest cumulative yield efficiency of 0.440–0.503 kg/cm^2^ with respect to the trunk cross-section area (TCSA). The least productive cultivar ‘Vega’ produced fruits of the highest average weight—9.9 g. Fruits of ‘Regina’ and ‘Sunburst’ were large as well—8.8–9.1 g. ‘Irema BS’ fruits had the highest soluble solids content (SSC)—20.2%. The lowest SSC was recorded in ‘Merchant’ and ‘Sunburst’ fruits—14.7–15.8%. The yield of advanced selection, No. 102, equaled to the yield of cv. ‘Regina’. No. 102 had a high fruit weight, and fruits were distinguished by attractiveness and taste.

## 1. Introduction

Global sweet cherry production exhibited slight increasing tendencies in recent years [1] and reached about 2.5 million tons per year [2]. Northeastern European countries are far from being the top producers of sweet cherries. In Lithuania, cherry orchards comprise only about 4% of the total plot under fruit and berry plants [3]. Sweet cherry is mainly grown by amateur growers, and they are very rarely planted in commercial orchards. Growing sweet cherry is more complicated in more severe climatic conditions. Low winter temperatures may damage flower buds or the entire fruit tree [4,5]. Recently, winters have become milder due to climate change, but fluctuations in temperatures create other challenges for fruit trees. Spring frosts also introduce a particular risk for early blooming sweet cherry trees [6].

In northern regions, the selection of optimal sites for orchard establishment and proper cultivar choice is crucial for successful sweet cherry growing. In Norway, sweet cherries can be grown up to 60° N latitude in areas with suitable local climatic conditions [7]. Lithuania is located between 53.53° and 56.27° N latitudes and has a coastline with the Baltic Sea, which softens winters mainly in the coastal area. For successful fruit production, sites of higher locations and with more favorable microclimates are also significant [8].

The focus of breeding new sweet cherry cultivars is on fruit quality, high and stable fruit set and tolerance to biotic and abiotic stress [9]. Recently, the weight of fruits available on the market reached 10 grams or more [10]. The fruit market has become global, but fruit growers face various challenges related to the specifics of a particular area. Fruit tree winter hardiness and tolerance to spring frost are of great importance in northern climates. Eighteen cultivars of sweet cherries were created in Lithuania from 1986 to 2022 [11]. The fruit weight of the first Lithuanian cultivars is 4–5.5 g [12,13] and does not meet the expectations of modern markets. Thus, efforts are being made to breed cultivars with large, tasty fruits during different ripening times [14]. Newly developed cultivars such as ‘Germa’, ‘Lukė’, ‘Irema BS’, ‘Pagunda’ and some others are characterized by better fruit quality indicators and sufficiently good adaptability to local climatic conditions [11]. Sweet cherry breeding programs are also being developed in Latvia and Estonia—countries located to the north of Lithuania [15,16]. Breeding and cultivar evaluation is a routine process in expanding the knowledge and capabilities of sweet cherry growing [17,18,19,20]. Cultivars originated in neighboring countries, Western and South Europe, USA, Canada, Ukraine and Russia are mainly found in Lithuanian orchards. Promising cultivars for testing usually are selected based on the experience of other researchers or personal observations.

Vigorous trees on *Prunus mahaleb* L. and *P. avium* L. seedlings are still used in sweet cherry production in Lithuania [21]. It is appropriate to conduct research with dwarf fruit trees used in modern orchards [22]. Gisela 5 usually meets these requirements [23,24,25]. It also proved to be suitable as growth limiting and yield-efficient rootstock in northern conditions [26,27,28]. Gisela series rootstocks are highly adaptable and can be found globally [29]. There is information that Gisela 5 rootstock in northern regions performs even better than in southern ones [30].

The study aims to investigate the main characteristics of fruit tree growth, productivity and fruit quality of sweet cherry cultivars and advanced selections grafted on dwarf rootstock Gisela 5.

## 2. Results

According to sweet cherry trunk diameters after 7 years of growth, ‘Mindaugė’ and ‘Vega’, also No. 102, were the most vigorous cultivars (Figure 1). Cv. ‘Merchant’ and No. 6 exhibited the weakest growth.

Sweet cherry trees began to flower in the second year after planting. The most abundant flowering was recorded for cv. ‘Regina’, and the lowest was recorded for ‘Irema BS’ with (Table 1). The fruit trees flowered less in the third leaf, but ‘Regina’ remained the most abundant flowering cultivar. Since 2018, flowering has become more abundant and stable. In different years, the cultivars that bloomed the most were ‘Regina’, ‘Sunburst’, ‘Mindaugė’, ‘Irema BS’, and No. 6. Fruit trees bloomed in 2020 in a particularly abundant manner. In the following year, the fruit trees of most cultivars bloomed less, especially ‘Vega’, due to unfavorable weather conditions during winter and early spring (Table 1). Low temperatures damaged flowers or even flower buds. The ‘Vega’ cultivar and ‘Sunburst’ and ‘Merchant’ were the most affected. (Table 2). Cvs. ‘Regina’, ‘Irema BS’ and No. 6 exhibited the best flower survival.

In the first two years, the yield was very low and was not recorded. From the fourth year onwards, the yield was determined. In 2018, ‘Mindaugė’ fruit trees were the most productive, and ‘Merchant’ trees were the least productive (Table 3). The latter cultivar produced the most abundant yield in the next year, while ‘Regina’ produced the lowest yield. In the sixth leaf (2020), ‘Regina’ and No. 102 produced the highest yields, and ‘Vega’ produced the lowest. The year 2020 was characterized by the most abundant harvest during the entire period of research. ‘Irema BS’ was the most productive, and ‘Vega’ produced the lowest yield. However, the average yield was about twice lower in the next year. On average, cvs. ‘Regina’, ‘Irema BS’, and No. 102 produced the highest average yield during the study period, and ‘Vega’ was the least productive.

There were no statistical differences in yield efficiency among cultivars in 2018 (Table 4). Cv. ‘Merchant’ was the most, and ‘Regina’ exhibited the least yield efficient in 2019. In 2020, when the highest yield was recorded, the yield efficiency of most tested cultivars was similar—0.206–0.261 kg/cm^2^. ‘Regina’ showed the highest yield efficiency, and ‘Vega’ and ‘Mindaugė’ showed the lowest. The latter cultivar was the most yield-efficient in the last year of the experiment. ‘Vega’ fruit trees grew vigorously, produced a small yield and had the lowest cumulative yield efficiency. Cv. ‘Regina’ trees were the most yield-efficient followed by ‘Sunburst’, ‘Irema BS’ and ‘Merchant’.

Average fruit weight varied among cultivars and within the years. Fruits of cv. ‘Regina’ had the highest weight during the first harvest (Table 5), while fruits of ‘Mindaugė’ and ‘Irema BS’ were the smallest. From the second harvest onwards, cv. ‘Vega’ had the largest fruits. ‘Irema BS’ fruits in 2021 were the smallest during the study. According to the average data, the smallest fruits had Lithuanian cultivars and advanced selections, with the exception of No. 102. The least productive cv. ‘Vega’ had the largest fruits, followed by ‘Sunburst’ and ‘Regina’.

Fruit soluble solid content (SSC) varied from 11.7% in ‘Merchant’ fruits (2020) to 21.7% in ‘Irema BS’ fruits (2019) (Table 6). SSC in the fruits of the latter cultivar was steadily high throughout the study period and reached an average value of 20.2%. The lowest average SSC was recorded in ‘Merchant’ and ‘Sunburst’ fruits: 14.7 and 15.8%, respectively. A particularly low SSC was observed in 2020 when the fruit yield was the highest.

‘Merchant’, ‘Sunburst’, ‘Regina’ and advanced selection No. 102 fruits were the most attractive (Table 7). No. 6 fruits had a poorer appearance. ‘Sunburst’, ‘Mindaugė’, ‘Regina’ and No. 102 fruits were the most delicious, and ‘Irema BS’, ‘Merchant’, ‘Vega’ and No. 6 fruits were somewhat less tasty. ‘Sunburst’, ‘Regina’ and No. 102 fruits achieved the highest overall rating.

## 3. Discussion

The growth of fruit trees is determined by cultivar and rootstock [31,32]. Even with the same rootstock, fruit trees of different cultivars have different growth vigor [23,33]. The weaker growth of fruit trees is an advantage, especially in the case of sweet cherries. Based on our research, the most significant effort will be needed to limit the growth of ‘Mindaugė’, ‘Vega’ and advanced selection No. 102 fruit trees.

The early or late beginning of cropping can be determined by the genetic characteristics of the cultivar [34], but vigorous sweet cherry trees on Mazzard or mahaleb rootstocks often do not flower significantly until the sixth or seventh leaf [35]. Usually, dwarf rootstocks significantly increase fruit tree precocity [31,36]. In our experiment, sweet cherry trees on Gisela 5 began flowering already in the 2nd year. Similar studies reported the early onset of flowering fruit trees on Gisela 5 rootstock [37,38]. According to the flowering abundance, cv. ‘Regina’ was the most precocious. Despite the observation that all cultivars of fruit trees bloomed less in the next year, cv. ‘Regina’ exhibited the most abundant flowering.

During the first two years, the fruit set was relatively low, and yields were not recorded. A more significant harvest was achieved in the 4th year after planting. The yield of the ‘Merchant’ cultivar was the lowest, while ‘Mindaugė’ yielded almost twice as many—4.69 t/ha. In later years, the most productive cultivars were ‘Merchant’, ‘Regina’ and advanced selection No. 102. The highest yields were recorded in 2020. ‘Regina’ and No. 102 produced a yield that was slightly higher than 20 t/ha. Weather conditions during the winter–early spring period in 2019–2020 were the most favorable during the study’s period. Average monthly air temperatures were higher than perennial ones, and the minimum temperature was recorded in early spring when it fell only to 10.7 °C. Frosts falling down to −2.9 °C at the beginning of flowering could slightly damage earlier flowering cultivars. The next winter (2020–2021) was characterized by low average and minimum temperatures in January and February. Moreover, a large fluctuation in temperature was recorded in the January–March period. Long-term warming in the second half of winter break dormancy and subsequent cooling even with not very low temperatures, may damage plants. Salazar-Gutiérrez et al. [39] reported that hardiness is highly dependent on fluctuations in temperature. This phenomenon mostly affected ‘Sunburst’, ‘Merchant’ and especially ‘Vega’ cultivars. Damage to the flower buds of the latter cultivar was rated with 4.0 points, and the yield was almost lost. Cultivar ‘Mindaugė’ produced the largest yield after the unfavorable winter–early spring period, although some of its flowers were also damaged. The frost resistance of sweet cherry flower buds and flowers is a complex phenomenon. This is influenced by the critical values of temperatures and their change over a relatively long period.

According to Salazar-Gutierrez et al. [39], sweet cherry cold hardiness increases from late fall to mid-January and is later followed by deacclimation. The possible level of damage is very dependent on the stage of flower bud development, and warm conditions increase flower vulnerability while low temperatures decrease that condition. Another study established that flower bud damage during spring frosts depended on their stage of development [28]. Flower buds subjected to frost at an earlier stage are more susceptible. Sweet cherry flowering phenophases are advanced by warm weather prior to the onset of phenophase [40]. Therefore, the risk of frost will remain with climate change, causing variable yields.

The yield efficiency indicator describes the ratio of fruit tree yield and growth. Higher yield efficiencies indicate that the fruit tree produces more fruit rather than vegetative parts. As the weaker fruit tree grows and produces more fruit, its yield efficiency becomes higher. The low yield efficiency recorded for cv. ‘Vega’ was caused by vigorous tree growth and low yield. The most yield-efficient fruit trees were the fruit trees of cv. ‘Regina’ followed by ‘Sunburst’, ‘Irema BS’ and ‘Merchant’. In the study by Gjamovski et al. [10], ‘Regina’ had about twice higher cumulative yield efficiencies than ‘Sunburst’ at the fifth leaf.

It is genetically determined that sweet cherry fruit weights can vary greatly. In the fresh fruit market, cherries weighing at least 10 grams are preferred [41]. In our experiment, only cv. ‘Vega’ produced fruits of a similar size, but the yield of this cultivar was low. Fruit size is affected by crop load [42], so the fruit would likely be smaller at the higher crop level. ‘Sunburst’ and ‘Regina’ fruits weighed about 9 g. The average fruit weight of Lithuanian cultivars was almost 7 g, and only an advanced selection of No. 102 produced fruits weighing 7.6 g. In addition to genetics, fruit size is determined by local ecological conditions and the technological factors of orchard management. According to different studies, the weight of ‘Regina’ fruits when grown on Gisela 5 rootstock ranged from 8.3 [10] to 11.5 g [43] in different locations. The weight of fruits grown in northern countries is usually lower. Preference is given to fruit tree winter hardiness, and a fruit weight of 6–9 g is considered to be acceptable [15].

Another important indicator of fruit quality is the soluble solid content (SSC). It ranged from 14.7% in ‘Merchant’ to 20.2% in ‘Irema BS’ fruits. Many factors influence SSC, and its range is wide enough in sweet cherries. It may vary between 11 and 25 °Brix [44]. There is different information regarding the lower limits of the SSC range. According to Vangdal [45], in order to be of acceptable quality, sweet cherries in Norway should contain at least 14.2% of soluble solids. American consumers prefer ‘Brooks’ and ‘Bing’ fruits with an SSC of not less than 16.0% [46]. The lowest SSC in our study was found in the year of the most abundant yield. High crop loads may have certain negative effects on fruit SSCs [47], and reducing the crop load results in an increase in SSC [48].

Sweet cherries are mainly used for fresh consumption, so the attractiveness and flavor of fruits are very important characteristics. Almost all tested cultivars received high sensory ratings. The exception was No 6, which was rated significantly worse. Meanwhile, another advanced selection No 102 was rated equally to the best commercial cultivars. A relationship between fruit flavor and SSC is often discussed. Some studies show a positive relationship between these indicators [49,50]. Crisosto et al. [51] reported interesting data that reflected no differences in SSC and fruit flavor perception between trained panelists, but consumers preferred fruits with higher SSCs. A curvilinear relationship between SSC’s impact on flavor was established [52], which suggests that lower SSC values may limit flavor acceptance and high ones are not determinative. Relatively low SSC values in 2020 did not reflect negatively on the sensory evaluation of fruits. On the other hand, a high SSC in ‘Regina’ fruits did not produce a high taste score.

Cvs. ‘Sunburst’, ‘Regina’ ‘Vega’ and ‘Merchant’ should be the most valuable from a consumer’s point of view due to the good quality of the fruit. The high yield of fruit trees is also important for fruit growers. ‘Regina’ and ‘Sunburst’ performed best at a young age. The advanced selection No. 102 produced a yield equal to that of the ‘Regina’ cultivar. The average fruit weight of No. 102 was the highest among all studied Lithuanian cultivars and was equal to ‘Merchant’ fruits. The fruits of No. 102 were attractive with good flavor and were rated similarly to those of ‘Regina’, ‘Sunburst’ ‘Merchant’ or ‘Vega’. ‘Regina’, ‘Sunburst’ and No. 102 performed best both in terms of productivity and fruit quality.

## 4. Materials and Methods

### 4.1. Planting Material and Orchard Maintenance

Sweet cherry cultivars and advanced selections of different ripening times were selected (Table 8). There was a preference toward large fruit size and dark fruit color (only ‘Vega’ fruits are colored red-yellow).

Fruit trees on the Gisela 5 rootstock were planted in the spring of 2015, spacing them at distances of 4.5 × 2.5 m. Trees were trained as spindles, and orchard floor management included frequently mown grass in the alleyways with herbicide strips along tree rows. Ammonium nitrate and potassium sulphate were applied to herbicide strips in spring annually since the third leaf. The N_50_K_80_ rate was used.

### 4.2. Site and Meteorological Conditions

The field experiment was carried out in 2015–2021 at the Institute of Horticulture, Lithuanian Research Centre for Agriculture and Forestry (55°60′ N, 23°48′ E).

The soil at the experiment place was Epicalcari–Endohypogleic cambisol, containing 245 mg/kg of P_2_O_5_, 196 mg/kg of K_2_O, and 2.9% of humus, with a pH(1MKCl) of 7.2.

The minimum air temperature in February and March 2018 was −23.3 and −21.0 °C, respectively (Table 9). No frosts were recorded during flowering. The minimum temperature of −20.1 °C in the winter of 2018–2019 was recorded in January. On May 4–8 2019, spring frosts down to −3.2 °C occurred during the flowering of sweet cherry trees. The winter of 2019–2020 was mild. The lowest temperature of −10.7 °C was recorded in early spring. At the beginning of flowering, frosts down to −2.9 °C occurred on April 26. Low temperatures were recorded in January and February 2021, reaching −27.7 and −24.9 °C, respectively. A considerable fluctuation in temperatures was recorded in the January–March period. In 2021, fruit trees bloomed late (on 8–24 of May), and there were no frosts during the flowering period. Temperature extremes of the winter–spring period were not reached during the research period.

### 4.3. Observations and Measurements

Trunk diameter measurements were performed in autumn at 25 cm above the graft union. Fruit tree flowering abundance was recorded at full bloom and scored on a 0–5 score scale, where “0” means the lowest and “5” means the highest value. In 2021, flower damage induced by unfavorable temperatures during the winter period was evaluated on a 0–5 score scale, where “0” means the lowest and “5” means the most significant damage. The yield was recorded for the entire experimental plot and recalculated to t/ha. Annual yield efficiency was expressed as the ratio of fruit yield per tree (kg) and tree trunk cross-section area (TCSA, cm^2^). The cumulative yield efficiency was calculated as a sum of annual efficiencies. The average fruit weight (g) was determined on a representative sample of 100 fruits per experimental plot. The soluble solid content (SSC, % Brix) was established with a digital refractometer (ATAGO 101, Atago Co., Ltd., Tokyo, Japan) on juice extracted from a random sample of 20 fruits. Sensory fruit quality was examined at harvest by 7 trained panelists. Fruit appearance, flavor and overall rating were expressed on a 1–5 score scale, where 1 indicates the lowest quality and 5 is the highest. Sensory fruit quality evaluation was performed in 2018 and 2020.

### 4.4. Experiment Design and Statistical Analysis

The field experiment was arranged in four replications with randomly arranged cultivars, and each experimental plot included 3 fruit trees. Experimental data were subjected to a one-way analysis of variance (ANOVA), and their means were compared using Duncan’s multiple range test. Differences were considered to be significant at *p* ≤ 0.05.

## Figures and Tables

**Figure 1 plants-12-00614-f001:**
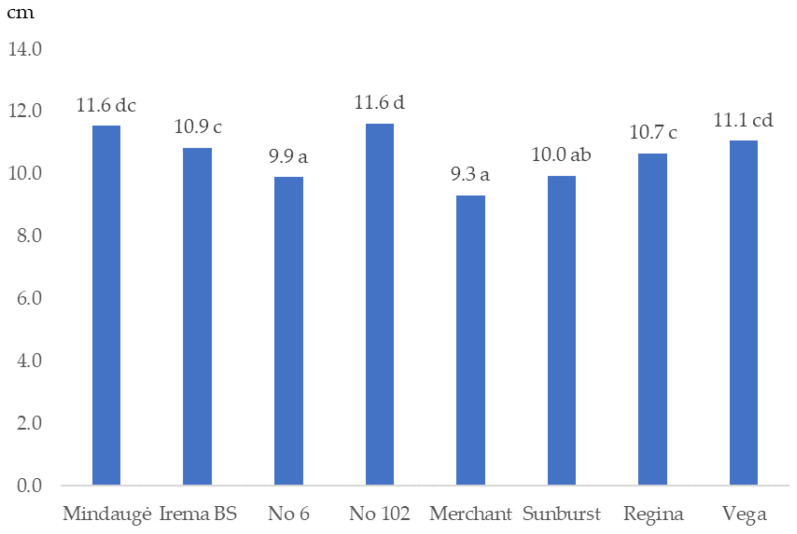
Sweet cherry tree trunk diameter in autumn 2021 in cm. Values marked with different letters are statistically different (*p* ≤ 0.05).

**Table 1 plants-12-00614-t001:** Sweet cherry flowering abundance (0–5 score scale). Values marked with different letters are statistically different within the columns (*p* ≤ 0.05).

Cultivar	2016	2017	2018	2019	2020	2021
‘Mindaugė’	2.0 bc	0.1 a	3.8 b	4.4 bc	5.0 b	4.4 cd
‘Irema BS’	0.3 a	0.3 ab	3.8 b	3.9 a	5.0 b	5.0 d
No 6	2.1 bc	1.4 bc	3.5 ab	4.1 ab	4.5 a	3.2 b
No 102	2.3 bc	0.8 b	3.6 ab	4.8 c	5.0 b	3.9 c
‘Merchant’	1.7 b	0.2 a	3.3 a	4.1 ab	4.7 ab	4.0 cd
‘Sunburst’	1.7 b	0.0 a	3.4 a	4.4 bc	5.0 b	3.7 bc
‘Regina’	3.4 c	2.3 c	4.2 c	4.7 bc	4.9 b	5.0 d
‘Vega’	1.8 b	0.0 a	3.8 b	4.2 ab	5.0 b	1.8 a

**Table 2 plants-12-00614-t002:** Sweet cherry flower damage after winter 2021 (0–5 score scale). Values marked with different letters are statistically different within the columns (*p* ≤ 0.05).

‘Mindaugė’	1.1 b
‘Irema BS’	0.5 ab
No 6	0.5 ab
No 102	1.0 b
‘Merchant’	1.8 c
‘Sunburst’	2.2 c
‘Regina’	0.0 a
‘Vega’	4.0 d

**Table 3 plants-12-00614-t003:** Sweet cherry fruit yields (t/ha). Values marked with different letters are statistically different within the columns (*p* ≤ 0.05).

Cultivar	2018	2019	2020	2021	Average
‘Mindaugė’	4.69 c	2.14 abcd	9.77 b	12.36 e	7.24 c
‘Irema BS’	3.58 abc	2.79 abcd	16.68 c	8.98 d	8.01 cd
No. 6	3.51 abc	2.69 abcd	12.76 b	5.59 b	6.14 bc
No. 102	2.48 ab	3.12 dc	21.20 d	6.80 bc	8.40 cd
‘Merchant’	2.42 a	3.33 d	12.21 b	5.06 b	5.76 b
‘Sunburst’	3.54 abc	3.05 bcd	16.25 bc	5.61 b	7.11 bc
‘Regina’	2.99 abc	1.69 a	22.16 d	8.07 cd	8.73 d
‘Vega’	2.88 abc	1.90 ab	5.85 a	0.37 a	2.75 a

**Table 4 plants-12-00614-t004:** Sweet cherry yield efficiency (kg/cm^2^ TCSA). Values marked with different letters are statistically different within the columns (*p* ≤ 0.05); ns—non-significant.

Cultivar	2018	2019	2020	2021	Cumulative
‘Mindaugė’	0.081	0.030 ab	0.114 a	0.135 e	0.359 b
‘Irema BS’	0.071	0.044 ab	0.224 bc	0.109 d	0.448 cd
No. 6	0.078	0.049 b	0.206 b	0.082 bc	0.415 c
No. 102	0.045	0.041 ab	0.252 bc	0.072 b	0.409 bc
‘Merchant’	0.063	0.071 c	0.224 bc	0.083 bc	0.440 cd
‘Sunburst’	0.080	0.057 bc	0.261 bcd	0.081 bc	0.478 cd
‘Regina’	0.060	0.026 a	0.315 d	0.102 cd	0.503 d
‘Vega’	0.059	0.031 ab	0.079 a	0.004 a	0.173 a
	ns				

**Table 5 plants-12-00614-t005:** The average sweet cherry fruit weight (g). Values marked with different small letters are statistically different within the columns and with the capitals—in the row (*p* ≤ 0.05).

Cultivar	2018	2019	2020	2021	Average
‘Mindaugė’	6.1 a	7.1 ab	7.8 ab	6.5 b	6.9 a
‘Irema BS’	6.5 ab	7.2 b	7.2 ab	5.7 a	6.7 a
No. 6	7.0 b	6.6 a	6.9 a	6.2 ab	6.7 a
No. 102	7.4 b	8.3 cd	7.5 ab	7.2 b	7.6 b
‘Merchant’	8.1 c	8.1 c	8.0 b	8.0 c	8.0 b
‘Sunburst’	8.9 d	10.2 e	8.1 b	9.2 d	9.1 c
‘Regina’	9.5 e	9.6 d	8.9 b	7.2 b	8.8 c
‘Vega’	7.5 b	11.6 f	10.2 c	10.4 e	9.9 d
Average	7.6 A	8.6 C	8.1 B	7.6 A	8.0

**Table 6 plants-12-00614-t006:** Soluble solid content of sweet cherry fruits (%). Values marked with different small letters are statistically different within the columns and with the capitals—in the row (*p* ≤ 0.05).

Cultivar	2018	2019	2020	2021	Average
‘Mindaugė’	19.0 b	20.4 c	15.7 c	15.2 a	17.6 d
‘Irema BS’	20.3 c	21.7 d	19.4 d	19.5 c	20.2 f
No 6	20.8 c	19.8 c	15.6 c	18.3 bc	18.6 e
No 102	19.2 b	20.2 c	13.6 b	14.0 a	16.8 c
‘Merchant’	16.7 a	16.0 a	11.7 a	14.3 a	14.7 a
‘Sunburst’	17.0 a	18.4 b	12.3 a	15.3 a	15.8 b
‘Regina’	17.9 ab	18.5 b	15.5 c	17.0 b	17.2 cd
‘Vega’	17.2 a	21.0 cd	14.5 bc	18.7 c	17.8 d
Average	18.5 C	19.5 D	14.8 A	16.5 B	17.3

**Table 7 plants-12-00614-t007:** Sensory evaluation of sweet cherry fruits (1–5 score scale), and average data of 2018 and 2020. Values marked with different letters are statistically different within the columns (*p* ≤ 0.05).

Cultivar	Appearance	Flavour	Overall Rating
‘Mindaugė’	4.5 ab	4.6 ab	4.5 ab
‘Irema BS’	4.5 ab	4.4 a	4.5 ab
No 6	4.4 a	4.4 a	4.4 a
No 102	4.7 b	4.6 ab	4.7 b
‘Merchant’	4.7 b	4.5 a	4.6 ab
‘Sunburst’	4.7 b	4.7 b	4.7 b
‘Regina’	4.7 b	4.6 ab	4.7 b
‘Vega’	4.6 ab	4.5 a	4.6 ab

**Table 8 plants-12-00614-t008:** Genetic resources [11,53]. OP—open pollination.

Cultivar	Fruit Ripening	Origin	Pedigree
‘Mindaugė’	midseason	Lithuania	‘Severnaya’ × ‘Jurgita’
‘Irema BS’	late	Lithuania	No 1106 × ‘Sam’
No. 6	midseason	Lithuania	unknown
No. 102	midseason	Lithuania	‘Ulster’ × OP
‘Merchant’	early	Great Britain	‘Merton Glory’ × OP
‘Sunburst’	midseason	Canada	‘Van’ × ‘Stella’
‘Regina’	late	Germany	‘Schneiders’ × ‘Rube’
‘Vega’	midseason	Canada	‘Bing’ × ‘Victor’

**Table 9 plants-12-00614-t009:** Air temperature in the winter–spring period. Records of the iMETOS^®^ (Pessl Instruments GmbH, Weiz, Austria) meteorological station in Babtai and Lithuanian Hydrometeorological Service [54] (* Galerija—Meteo.lt; ** Extreme phenomena—Meteo.lt).

Temperature, °C	December	January	February	March	April	May
	2017–2018					
Average	1.3	−1.4	−5.6	−0.4	10.4	16.8
Minimum	−4.2	−17.2	−23.3	−21.0	−1.5	2.1
Maximum	8.4	7.7	3.2	9.0	27.4	31.6
	2018–2019					
Average	−0.3	−3.6	1.6	3.4	8.2	11.7
Minimum	−13.6	−20.1	−8.8	−5.6	−4.2	−3.2
Maximum	5.6	5.2	10.3	13.9	26.1	28.5
	2019–2020					
Average	2.6	2.9	2.5	3.6	6.8	10.4
Minimum	−7.3	−3.9	−7.0	−10.7	−5.2	−2.2
Maximum	10.3	7.5	10.7	17.7	21.6	22.9
	2020–2021					
Average	0.8	−3.6	−5.9	1.9	6.0	11.5
Minimum	−9.4	−27.7	−24.9	−11.2	−3.8	−1.8
Maximum	6.3	5.0	13.6	16.7	21.0	26.2
	Perennial (1991–2020) average and extreme phenomena					
Average *	−1.2	−3.0	−2.4	1.2	7.6	13.0
Minimum **	−34.0	−40.5	−42.9	−37.5	−23.0	−6.8
Maximum **	15.6	12.6	16.5	21.8	31.0	34.0

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
