# Peer review of "Performance of Sweet Cherry Cultivars and Advanced Selections on Gisela 5 Rootstock in Young Orchards"

_plants, 2023, doi:10.3390/plants12030614_

Round 1
Reviewer 1 Report
The paper described a rootstock test for sweet cherry growing in Lithuania. Unfortunately, only one test plot with different cultivars grafted on the rootstock Gisela 5 at one location were studied. No traditional cultivars and rootstocks were included in the test design. Therefore, no comparisons can be made with the existing sweet cherry growing in Lithuania. Also results or information to the common sweet cherries in Lithuania were not described or discussed in the introduction and discussion. Therefore, the value of the results held up is not comprehensible.
Some comments/questions
Information to sweet cherry characteristics important for the fruit grower should intruded in the introduction and discussed in the discussion.
What are the most important fruit characteristics for the fruit growers and the marked in Lithuania?
What was the reason to use the selected sweet cherry cultivars and the rootstock Gisela 5?
What sweet cherry cultivars and rootstocks were used in Lithuania currently?
Was the time period of the investigations sufficient for a recommendation for fruit growing? Normally, cultivation trials need 8 to 10 years.
In a revised form of the manuscript, the most important results for the cherry growing in Lithuania should be highlighted and discussed.
The English should also be improved also.
Reviewer 2 Report
The manuscript “ Sweet cherry cultivar and selected number performance on Gisela 5 rootstock in the young orchard” seems to be an interesting topic. In this study the authors have tested the performance of 6 sweet cherry cultivars and 2 selected numbers on Gisela 5 rootstock. After careful evaluation I suggest including some information for which I recommend for minor revision.
Line 71-74: Please write the information more specific, particularly which parameters of the environment may have effect on plants.
Table 4: In case of 2018 the lettering has no value. Instead it’s better to mention it as non-significant.
Statistical information should be written in a different sub-heading with details information.
Reviewer 3 Report
This manuscript had evaluated performances of selected eight sweet cherry cultivars from 2015-2021. Authors ranked cultivars based on statistic analysis of phenotypic traits, such as yield and fruit quality traits. Although the manuscript could be interesting for sweet cherry growers in Lithuanian, the mechanism underlying the phenotypic variations is needed. Genetic background of these cultivars should be provided. In the M&M, I suggest authors add sub-titles: plant material, phenotypic measurements, experiment design, and statistical analysis. It could be easier to understand the manuscript in this way. In Result section, phenotypic traits of cultivars were ranked in different years. A combined analysis should be used to understand the effect of cultivar by year interaction.
Round 2
Reviewer 1 Report
With the additional information and changes, the manuscript can be recommended for publication. The cultivation of sweet cherries has not yet achieved great importance in Latvia. The present results can make a contribution to a modern and successful sweet cherry cultivation in Latvia.
Reviewer 3 Report
Authors addressed my comments in the manuscript. I think the paper is now acceptable for publication.